# Resilience to abrupt global catastrophic risks disrupting trade: Combining urban and near-urban agriculture in a quantified case study of a globally median-sized city

**Matt Boyd**[1]*, **Nick Wilson**[2]

**1** Adapt Research Ltd, Reefton, New Zealand, **2** Department of Public Health, University of Otago, Wellington, New Zealand.

☯ These authors contributed equally to the work.
* matt@adaptresearchwriting.com

## Abstract

### Background

Abrupt global catastrophic risks (GCRs) are not improbable and could massively disrupt global trade leading to shortages of critical commodities, such as liquid fuels, upon which industrial food production, processing and distribution depends. Previous studies have suggested urban agriculture as a resilience measure in the context of climate change and other natural hazards.

### Aims

To estimate the contribution a radical pivot to urban agriculture could have in building resilience to GCRs and the near-urban industrial agriculture needed to supplement urban food production.

### Methods

We determined optimum crops through mathematical optimization for food calorie and protein supply per land area for both urban and near-urban (industrial) agriculture. We calculated the land area available for food production within a temperate globally median-sized city using Google Earth image analysis of residential lots and open city spaces. We calculated the population that could be fed through urban agriculture alone, and the extra near-urban land required for cropping with industrial agriculture to feed the remaining city population, under both normal climate, and potential nuclear winter conditions.

### Results

The optimal crops for urban agriculture were peas (normal climate), and sugar beet/ spinach (nuclear winter); while those optimal for industrial near-urban production were potatoes (normal climate), and wheat/carrots (nuclear winter). Urban agriculture could feed a fifth (20%) of the population. At least 1140 hectares of near-urban cultivation could

**Data availability statement:** Data were extracted from Google Earth, and other publicly available sources as detailed in the manuscript and presented in the Supporting Information. Spreadsheets of calculations are available at the following url: https://adaptresearchwriting. com/wp-content/uploads/2024/11/241107-data-calculations-urban-near-urban-agri.zip

**Funding:** The study was self-funded by the authors and no external funder had any additional role in the study design, data collection and analysis, decision to publish, or preparation of the manuscript. The specific roles of these authors are articulated in the 'author contributions' section.

**Competing interests:** No support was received from any organisation for the submitted work beyond the authors primary affiliations; the authors declare no financial relationships with any organisations that might have an interest in the submitted work; and no other relationships or activities that could appear to have influenced the submitted work. MB is the owner and sole employee of Adapt Research Ltd, this does not alter our adherence to PLOS ONE policies on sharing data and materials.

make up the shortfall. Another 110 hectares of biofuel feedstock like canola (rapeseed) could provide biodiesel to run agricultural machinery without fuel trade. Significantly more cultivated area is needed in nuclear winter scenarios due to reduced yields.

## Conclusion

Relatively little optimized near-urban industrial agriculture, along with intensified urban agriculture could feed a median-sized city in a GCR, while minimizing fuel requirements. Governments and municipal authorities could consider land use policy that encourages development of urban agriculture and near-urban cultivation of optimal crops, along with processing and local biofuel refining capacity.

## Introduction

### Global catastrophic risks and food security

Abrupt global catastrophic risks (GCRs) include nuclear war, extreme pandemics, technological catastrophes, massive volcanic eruptions, asteroid/comet impacts, or solar storms, among others [1]. Collectively these risks are not improbable [2,3] and some risks may be increasing (e.g., those associated with artificial intelligence, bioengineered pandemics and nuclear war). A GCR could cause immense disruption to global trade [4–6]. Such major trade disruption could lead to shortages of critical commodities, such as liquid fuels, necessary for industrial food production and distribution. Degradation of food production or distribution systems could lead to famine, whether impacting food production [7], or supply chains [8], and famine is a key historical driver of societal turmoil and civilization collapse [9,10].

Even if measures have been taken to secure liquid fuel supply, for example upscaling biofuel feedstock production and refining capacity [11], then shortages of industrial agricultural inputs might still threaten famine [7], and the two-thirds of countries that are net food importers [12], would be at risk, as would remote isolated nations with potentially fragile linkages to global shipping routes [4].

One possible way to mitigate the impact of a severe disruption to global trade, and thereby prevent famine and/or societal collapse, is to pivot food production to more fuel-efficient, low input methods, closer to where food is consumed. Urban agriculture (UA) is the closest and most fuel-efficient source of food for city dwellers and may be part of one such solution, with some amount of near-urban land needing to be farmed as well.

### Urban agriculture

UA can be defined as, "an industry located within a town, a city or metropolis, which grows or raises, processes and distributes a diversity of food and non-food products (re-)using largely human and material resources, products and services found in and around that urban area, and in turn supplying human and material resources, products and services largely to that urban area" [13]. UA may include such entities as home gardens, community gardens, allotments, school gardens, balcony and rooftop gardens, or larger farms at urban/rural interface [14], as well as indoor farms, vertical farms, and building integrated agriculture [15]. These can be conceived as 'micro-UA' (e.g., home gardens), 'meso-UA' (e.g., community gardens), or near-urban farms on the urban perimeter [16]. In this paper we use the term 'near-urban' agriculture to include industrial agriculture that is close to the edge of the built urban environment (i.e., where transport fuel needs are very low).

UA offers many potential benefits, including the provision of nutritious food, alleviation of poverty/food insecurity, strengthened communities, inclusion of marginalized groups, food security in the face of climate change, reduced 'food miles' (i.e., greenhouse gas emissions due to transport of food) [13], heat island reduction, employment, biodiversity, urban water retention, ecosystem services, circularity (reuse of waste and regeneration of inputs), dietary variety and progress towards the Sustainable Development Goals [15]. UA also contributes to education, 'urban green infrastructure' and the functioning of natural cycles disrupted by urban sprawl [14].

On the other hand, barriers to UA include potential contamination of urban soils [13,17], shortages of water supply, sunlight shading due to the built urban environment, or the economics of food production versus higher rent functions [14].

## Previous research on urban agriculture

Previous studies have shown that UA makes a significant contribution to global food supply [18], particularly vegetable supply [19,20], with production area of home gardens potentially threefold that of community gardens [21]. Table 1 describes a selection of studies that report the potential of UA in terms of human diet. Previous findings indicated that vegetable

**Table 1. Examples of studies that estimate the potential contribution of urban agriculture.**

| Author/s | Setting | Details | Quantitative result | Comment |
|---|---|---|---|---|
| Hume, Summers, & Cavagnaro, 2021 [28] | Adelaide, Australia | High resolution optical imagery to quantify residential land immediately available for vegetable production (i.e., residential lawn) in a low-density city. | Under the 'medium yield' scenario 72% of lawn area required to meet recommended vegetable intake. Mean yield 5.08 kg/m²/per annum. | Only considered fruit and vegetables (F&V) intake not food energy. Considered 'hypothetical' commercial yield based on arbitrary mix of crops. |
| Grafius et al., 2020 [20] | UK towns of Bedford, Luton, and Milton Keynes | Estimated potential food production from land currently used for some form of food production: including allotment sites, private residential gardens and urban fruit trees. | Current food production estimated to supply urban population with F&V for about 30 days per year (i.e., 8% of F&V requirements). | Only current production, not potential production in catastrophe. |
| Taylor & Lovell, 2012 [21] | Chicago, Illinois | Included residential gardens, vacant lots, community gardens, urban farms, and school gardens | Does not specify how much food is or could be grown | Used Google Earth data and could be easily replicated in other cities. |
| McClintock, Cooper, & Khandeshi, 2013 [29] | Oakland, California | Identified vacant lots, open space, and underutilized parks with agricultural potential. | 1,200 acres (public sites) and 864 acres (private sites). Public land could support 2.9–33% and private land 2.1–24.5% of current vegetable consumption. | Calculations by crop weight rather than food energy per capita. Reported intake far below recommended vegetable intake. |
| Saga & Eckelman, 2017 [19] | Boston, Massachusetts | Estimated UA potential, including rooftop and ground level areas. | Mapped areas have potential to produce enough fresh F&V for approximately one million people (50% larger than Boston's population). | Rooftops may be difficult to prepare in a catastrophe; also considered F&V intake, not food energy. |
| Napawan & Burke, 2016 [30] | San Francisco Bay Area, California | Potential for existing residential lots to provide caloric needs of residents in four case study communities. | Residential lots could provide F&V requirements: San Francisco: 66%, Oakland: 104%, Dublin: 190%, Mountain House: 70%. | Only residential lots. Did not evaluate food energy potential. |
| McDougall et al., 2020 [24] | Sydney, Australia | Evaluated potential domestic yard space plus street verges and unused land. | Highest land use scenario, UA could supply all required calories to 901,966 people (33.8% of total population). | Analyzed sample of city with extrapolation; based yield on studies of actual crops not optimized cropping. Residential lots probably overestimated (total lot minus house footprint). |
| Munya, 2016 [16] | Auckland, New Zealand | Analysis of three 'representative' city blocks, one high, one medium, one lower density – extrapolated to whole city. | 25% of back and front yards of residential lots in low-density suburbs would contribute 9.4–12.3% to overall diet of urban residents. | Extrapolated from small case study blocks to entire cities. |

requirements of some urban populations could be met through UA, but UA falls short of meeting total food demands. Quantitative estimates of UA production are still relatively rare, and there is considerable potential to increase urban food production [20].

Most UA activity and research has focused on community gardens and green open spaces (rather than private gardens), or on residential lots alone, and also on issues of sustainability, poverty, and climate change [13], or regional disasters such as earthquakes [22]. Some have argued that UA focus should be on nutrients and nutrient security alone and that 'only partial self-sufficiency is required and desired' [23], rather than focus on potential abrupt global catastrophes, international supply chain risks and general resilience.

Some UA research exaggerates the impact of UA by applying the advantages of some growing practices to all potential UA [23]. Additionally, when residential lots have been considered, some research did not investigate residential properties as a stand-alone resource, and 'yards' were simplistically considered to be the total space of the residential block minus the building footprint [24], which likely includes many paved areas.

When considering resilience to GCRs, limitations of many previous studies include that they did not examine all three components of UA (micro-, meso-, and near-urban), or examined only existing UA land area rather than potential UA land area, or studied only actually grown foods rather than optimized combinations of foods, or did not report the offset in industrial agriculture (rural) that UA might entail. Much previous research on UA also focuses on large urban areas, rather than more representative median cities [25]. The median city population globally is 85,000 [26]. The estimated area required to satisfy global cereal (661.6 Mha) and vegetable (47.2 Mha) demand exceeds global urban land area (64.3 Mha) [25], but this does not consider the differing density and context of cities, such as small semi-rural cities.

## Food system resilience

Overall, what is needed is *food system resilience* and UA may be a key component in the face of potential GCRs. Food system resilience can be defined as the "capacity over time of a food system and its units at multiple levels, to provide sufficient, appropriate, and accessible food, to all, in the face of various and even unforeseen circumstances" [15]. A primary consideration is whether there is sufficient land to produce food. In a GCR that has severely degraded trade, fuel supply and transportation, productive land must be near to consumers.

Given this background, we aimed to determine the potential contribution that a radical pivot to UA might have in building resilience to GCRs, by calculating the potential land area available for UA in a globally median-sized city for population (the New Zealand city of Palmerston North). Our approach is unique in considering all the following factors in combination: micro-, meso-, and near-urban UA, a median-sized city, optimized rather than typical crops, food energy/protein needs – not just vegetable intake, maximum catastrophe-context land use, analysis of shortfall to be made up by near-urban agriculture, consideration of altered climate in a nuclear winter scenario, and the volume of liquid fuel conserved by offsetting off-road (on-farm) diesel use. We used a simple method that can be replicated by other small cities. Additional considerations beyond land area and optimal crops, such as water supply [16], fertilizer and organics [24], agrichemicals, and expertise [27] are addressed in the Discussion.

## Study aims

Overall, this study aimed to address the following interrelated research questions:

1.  What are the optimal crops to grow in the UA and near-urban spaces to maximize the number of people fed?

2. How much land is available for UA in this median-sized case-study city?

3. How many people could this combination of UA space and crop selection feed?

4. How much additional near-urban industrial cropping is needed to make up the shortfall of micro- and meso-UA and feed the whole city?

## Materials and methods

No human participants were involved in this study and the research used publicly available information. As such, the study was exempt from institutional review board requirements

### Study site

For this initial case study, we chose a globally median sized city by population in a temperate climate. Our method and study are demonstrative, and the approach could be replicated for cities of different sizes in different locations. The study site was the city of Palmerston North, New Zealand. Palmerston North lies in a temperate location (Cfb/oceanic climate, mean yearly temperature ~ 12 C, with ~ 1000mm annual rainfall), has a metropolitan administrative area of 395 square km (official city limits), and a contiguous built urban environment of approximately 35 km². The city had an urban population of 91,800 in 2023 [31], mostly living in medium density suburban-style standalone housing.

To investigate the potential role of UA in mitigating the impact of a trade disabling global catastrophe, we sought to answer the following questions about our case-study site.

### What are the optimal crops to grow in the UA and near-urban spaces?

Previous research has shown a large discrepancy in total land area required to satisfy population protein and food energy needs by type of crop grown. For example, industrial agriculture producing only milk would require 642 thousand hectares (ha) to feed the total New Zealand population, whereas wheat requires 117 thousand ha and potatoes just 84 thousand ha [11]. The same issue arises with UA. Therefore, in our analysis, unlike many previous UA studies, we optimized for potential protein and food energy yield.

For the near-urban land area, we used the same general optimization approach as in our previous published work on optimizing frost-resistant crops for industrial agriculture in post-catastrophe New Zealand [32]. In this previous work, we considered published crop yield (kg/tonne of harvested crop per land area) and actual dietary energy (kJ) and dietary protein consumption of average New Zealanders, based on published nutrition survey research (specifically 8686 kJ of dietary energy and 81 g of dietary protein per person per day for a year) [33]. The optimization involved linear programming conducted with Excel Solver using the "Solver LP" method. Linear programming was selected as an optimization method due to its computational efficiency and ability to effectively handle multiple constraints while maximizing a specific objective function, making it well-suited for agricultural resource allocation problems where we need to simultaneously optimize for both protein and energy requirements under land constraints. For the nuclear winter scenarios, we considered frost-resistant crops that are commercially grown in New Zealand [34]. A simplifying assumption was one crop cycle per year.

For the micro- and meso- levels of UA, we analyzed as above but used crop yield data for UA from a recent meta-analysis by Payen et al. [35]. We considered all crops where the mean yield was over 1 kg/m² (gross weight), where at least two study results were reported in the meta-analysis and used data for open-air soil-based agriculture (i.e., excluding: polytunnels, green houses, aquaculture, and indoor settings with hydroponics and artificial lighting).

Herbs, garlic and cereal crops were not included for the micro-/meso- UA analyses. The rationale for this was that herbs and garlic are typically used as condiments and unlikely to be a palatable diet on their own; and because cereal crops typically have low yields (<1 kg/ $m^2$) in UA settings according to Payen et al. and are more complex to process than vegetables and fruits. We calculated arithmetic means, as per the Payen et al. meta-analysis. We present specific tabulated data on yields and nutrient levels for these crops in the Supporting Information. These protein and energy levels were obtained from the food composition database of the New Zealand Institute for Plant and Food Research [36]. We additionally present yields reported by experienced permaculture growers for comparison [27], but excluded this from our main analysis as ordinary growers may lack the relevant skill set and results were extracted from a single detailed report.

## How much land is available for UA in this median-sized case-study city?

We defined the boundary of the city by constructing a Google Earth polygon around the built urban environment, where it typically abuts farmland (see Fig 1). Within this boundary we estimated land area potentially available for UA in the form of micro-UA (residential lots), and meso-UA (city green spaces). This was estimated through manual analysis of samples, using orthorectified Google Earth images.

**Micro-UA.** We used a random geolocation generator to identify specific residential lots within the built urban boundary (http://www.geomidpoint.com/random/), piloting 20 lots to establish inter-rater reliability (both authors) and then completing a sample of the entire urban area (one rater). Residential site area was divided into roof area, paved area, non-paved area, and tree area [16]. We considered only non-paved area as suitable for UA, identifying these areas using Google Earth imagery (data attribution 20 February 2024), a local real estate database (www.homes.co.nz) providing aerial imagery and property photographs, supplemented with Google Street View to confirm non-paved green space. We defined unpaved areas with UA potential using the polygon tool in Google Earth to calculate land area. We chose Google Earth because the spatial resolution of the aerial images is high and analysis required minimal training [21]. This analysis could easily be replicated by city officials or community groups in diverse locations. We sampled iteratively until we had estimated the land area potentially suitable for UA in a number of residential lots equal to the desired sample size (see 'Statistical Analysis' below). We then extrapolated the mean potential UA area of each lot to the wider city by simple multiplication of the mean potential UA area and the number of residential lots in the city (see Table 2). The method was piloted and refined in another New Zealand town, where physical site visits allowed comparison between aerial imagery and on the ground reality. This revealed that adding the use of Google Street View and the photographs on the real estate website (www.homes.co.nz), greatly enhanced the accuracy of identifying potential UA space. We recorded whole property land area and the 2021 capital valuation of each residential lot (www.homes.co.nz). Fig 1 illustrates coding of an example residential lot.

**Meso-UA.** For meso-level UA, Google Earth was used to estimate the number of spaces with UA potential for the entire land area of the defined built environment. We iteratively sampled random 1 km x 1 km sections of the city, using the polygon tool on Google Earth to sum the potential UA area of these spaces. We included open unpaved/unwooded space, and parks [29], as well as unpaved school grounds, community gardens, golf courses, and sports fields. We excluded building footprints, rooftops, construction sites, waterway banks, trees and wooded areas, cemeteries, as well as paved areas, berms and street verges. We measured spaces to the edge of the built-up city, as per [20]. We excluded any part of 1 km x 1 km squares that overlapped with other squares or the built urban boundary. This process was continued until we achieved the desired sample size (see 'Statistical Analysis' below).

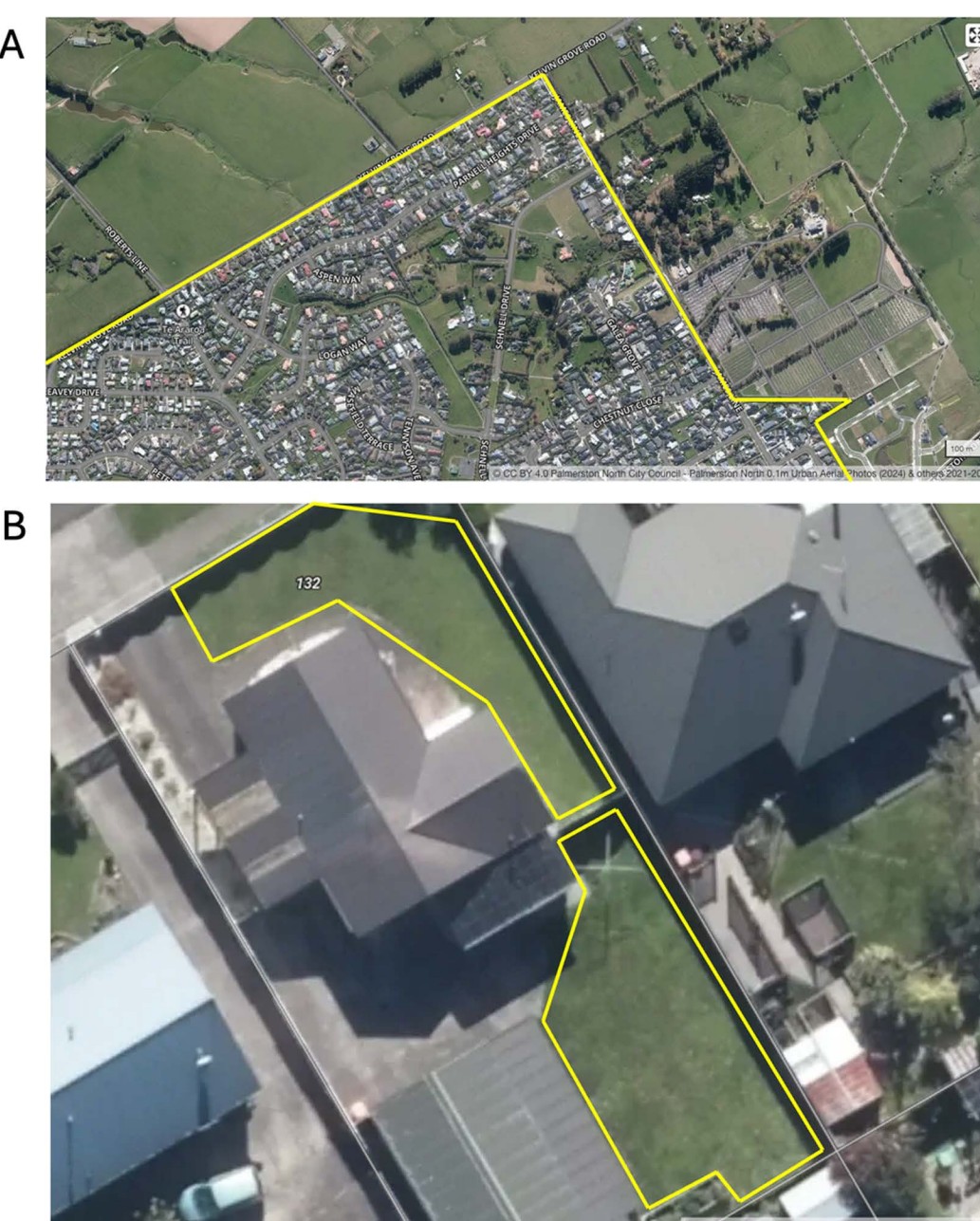

**Fig 1. Calculating land area for urban food production.** Publicly available aerial imagery (CC BY 4.0) from Palmerston North City Council (2024), similar though not identical to the views obtained for the study using Google Earth. (A) Detail of the built urban boundary showing clear demarcation between urban environment and surrounding near-urban land (cemetery to east of urban area was excluded, as although part of 'built' space it is unlikely to be used for agriculture; also excluded was new development at the eastern boundary, as houses were not yet built). (B) Detail of coding residential lots for potential urban agriculture (UA)-space. Note coding of front yard and back yard where there is clear open space (front yard sighted on Google Street View and polygon drawn around clear yard space). Note paved driveway excluded, as well as trees, hedges, and areas with uncertain UA potential (not shown, meso-UA area, where the same process used for yards was applied to city green spaces).

We multiplied the mean UA area of the sampled residential lots by the number of residential lots in the city. Similarly, the calculated UA area per km$^2$ was multiplied by the built urban area. The estimated total micro-UA space (i.e., total residential lots) plus meso-UA area (i.e., city open spaces) then defined the area potentially available for UA. We considered scenarios where 25%, 50%, and 75% of this potential space is used for UA, some space being needed for circulation of workers and storage of equipment/compost bins etc. Table 2 displays all variables, and their sources, used in the study calculations.

## How many people could this combination of UA space and crop selection feed?

We multiplied the total land area potentially available for UA by the protein and food energy (kJ) yield per area of the optimal crop(s) and determined what fraction of the total Palmerston North urban population's kJ and protein needs could be met if optimal food crop(s) were grown on the total potential UA land area with yields equal to those in the meta-analysis. We also considered scenarios where the yield of UA was reduced due to the sunlight blocking effects of a nuclear winter. Assuming both 150 Tg (equivalent to 150 megatonnes) or 5 Tg of soot has reached the stratosphere [37], with this previous study by Xia et al. specifically considering impacts on crop yields in New Zealand (with yield coefficients of 0.921 and 0.391 respectively, as in Table 2). Additional factors could also impact food production such as water supply, and inputs of fertilizer/organic material, agrichemicals and expertise. We address these additional factors in the Discussion.

## How much additional near-urban industrial cropping is needed to make up the shortfall of micro- and meso-UA?

Outside of the built urban area, near-urban cropping can take advantage of industrial methods to achieve high yields, while minimizing transport and fuel requirements to processing and consumption in the city. That said, different crops in different scenarios will require

**Table 2. Key variables considered in the analysis.**

| Variable | Value | Source or calculation |
|---|---|---|
| **Land available for urban agriculture (UA)** | | |
| Residential lots in the selected city: Palmerston North | 26,304 | New Zealand (NZ) Census 2018: https://figure.nz/chart/bU7j1ppye4qb4VYN |
| Total micro-UA area | Number of residential lots multiplied by mean area with UA potential per lot | Calculated from aerial imagery (see results) |
| Total meso-UA area | Area with meso-UA potential per km$^2$ multiplied by total built urban area | Calculated from aerial imagery (see results) |
| **Population food needs** | | |
| Population of Palmerston North | 91,800 | [31] |
| Food energy needs per capita | 8686 kJ/day | National NZ data [33] |
| Protein needs per capita | 81 g/day | National NZ data [33] |
| **Nuclear winter scenarios (these only consider frost-resistant crops)** | | |
| 5 Tg nuclear winter scenario | Yields multiplied by 0.921 | Reduction in yield specifically for NZ due to nuclear winter as per [37], and only considering frost-resistant crops |
| 150 Tg nuclear winter scenario | Yields multiplied by 0.391 | As per the row above. |
| **Liquid fuel for near-urban agriculture** | | |
| Biofuel yield (L) from canola crop | 1228 L/ha | NZ scenario [11] |
| Biodiesel requirement per ha | Wheat (43 L); Potatoes (118 L) | NZ scenario [11] |

different quantities of liquid fuel input to provide the expected yield. Additionally, in a GCR with trade collapse, usual supplies of imported diesel may not be available and local production of biofuel may be required [11]. Previous research has estimated the land area and diesel fuel requirements to feed the entire New Zealand population assuming any one of several different crops and documented industrial yield [11]. We used the results of these calculations for supplying minimum required food energy and protein, along with the diesel requirements per ha to calculate how much near-urban land would be needed to overcome shortfall in food energy and protein from micro- and meso-UA. We also calculated the land area needed for a biofuel feedstock such as canola (rapeseed) in case trade in liquid fuel is halted and local liquid fuel production is needed for agricultural machinery. We assumed zero transport fuel requirements, given that near-urban agriculture could be within walking distance of the city.

## Statistical considerations

With 26,304 standalone residential lots in Palmerston North, a sample of 96 lots was needed to achieve 95% confidence of a 10% margin of error. To ensure reliability, two researchers (both authors) iteratively discussed and trialed the method, and then independently calculated the potential UA area in a pilot of 20 randomly selected residential lots. Interrater agreement was high, with Pearson's r of 0.99, ICC 0.979, and a 2.4% difference in means (rater 1 mean 157.05 m$^2$, rater 2 mean 160.85 m$^2$). Further analysis by a single rater proceeded on this basis. A similar process was followed for meso-UA with two raters independently estimating the potential UA area in a randomly selected km$^2$ of the built urban area. There was a 6.0% difference in rater totals, and this was deemed acceptable. On the basis of the number of individual spaces coded in one km$^2$ (mean 22) it was estimated there would be 720 potential UA spaces across the city, with sample size of 85 needed to achieve 95% confidence with a 10% margin of error. One rater proceeded to estimate the potential UA area in random km$^2$, as above, until this number was reached. All calculations were performed using Microsoft Excel for Mac v16.9.

## Results

### Research Q1: Optimized cropping

For micro- and meso-UA areas under normal climate conditions, the optimized crop based on UA yields from the meta-analysis used was peas (at 292 m$^2$ of land to satisfy one person's dietary energy and protein needs per year) (Fig 2). If peas were excluded from the analysis, the next optimized crops were a mix of sugar beet and spinach. In the nuclear winter scenario, the optimized micro- and meso- area frost-resistant crops were a mix of sugar beet (326 m$^2$) plus spinach (18 m$^2$). If those two crops were excluded from the analysis, the next optimized frost-resistant crop was turnips (417 m$^2$) plus chicory (316 m$^2$).

For near-urban areas under normal climate conditions, the optimized crop based on commercial agriculture yields was potatoes (at 156 m$^2$, Fig 2). The next optimized crops were a mix of wheat plus carrots. In the nuclear winter scenario, the optimized near-urban frost-resistant crops were a mix of wheat (219 m$^2$) plus carrots (7 m$^2$) (as found for previously published work [32]). The next optimized frost-resistant crop was sugar beet (276 m$^2$).

### Research Q2: Potential micro- and meso-UA land area

The land area of the Palmerston North built urban environment was estimated to be 34.2 km$^2$ (34,181,997 m$^2$). The 96 sampled residential lots had a mean capital valuation (2021) of NZ$782,263 (US$480,539; 12 June 2024) (sd NZ$277,440). The mean lot area was 739 m$^2$ (sd 299.2). The mean area per lot potentially available for UA was 149.8 m$^2$, (sd 115.7; 95%

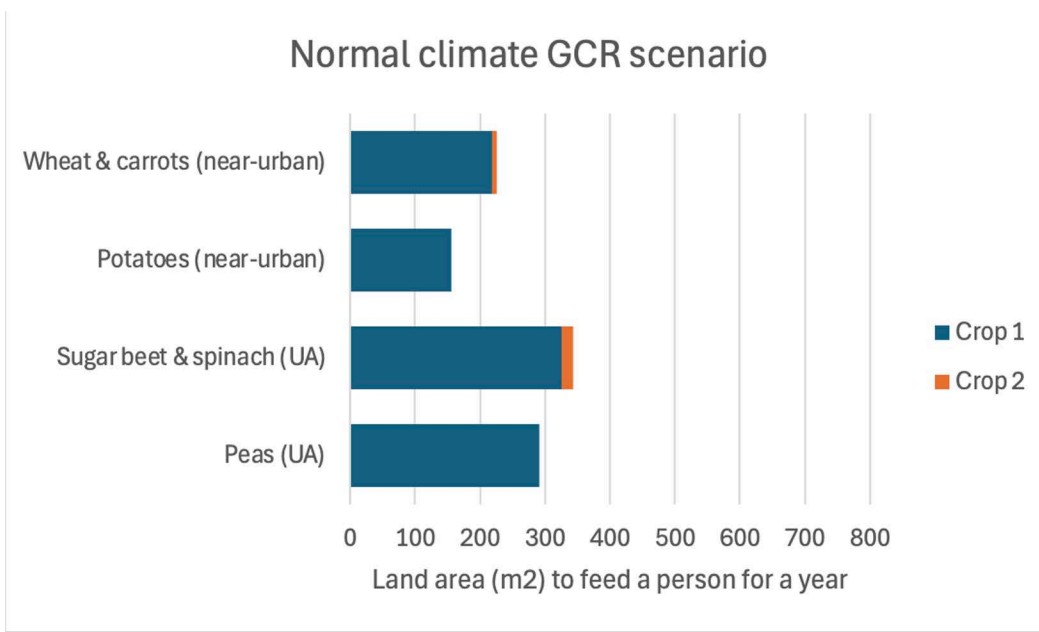

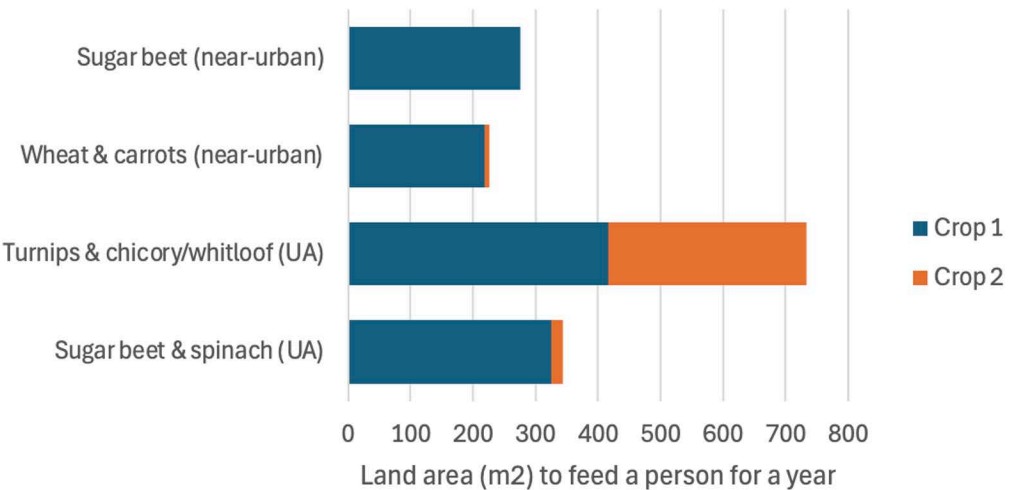

**Fig 2. Crop optimization results: Crops requiring least land area to feed one person - different agriculture types and post-catastrophe climate scenarios.** The figure shows the minimum land area required to feed one person in both the normal climate scenario (top panel) and nuclear winter scenario (bottom panel – only frost-resistant crops considered), whether deploying urban agriculture methods (UA) or near-urban industrial agricultural methods. Only the crops/crop combinations found to require the least and next least land area are displayed. Shorter bars in the graph indicate less land is required to provide protein and food energy needs (i.e., most optimal crops). Yields in the nuclear winter scenario are typical yields of frost-resistant crops and not adjusted here for reduced sunlight (see Table 3 for such an adjustment). GCR – global catastrophic risk; UA – urban agriculture.

CI: 126.7–172.9) which was 20.3% of overall lot area. With 26,304 'separate house' dwellings in Palmerston North, the result was 3.9 km² (3,940,339 m²) of potential micro-UA space in this city.

Coding of open spaces resulted in an estimate of 9.83% of total urban area having meso-UA potential (5,111,000 m² sampled from 34,181,997 m² built urban area), resulting in

an estimated 3.4 km² (3,360,090 m²) of potential meso-UA space. The resulting sum of micro- and meso- potential UA space for the whole city was 7.3 km² (7,300,429 m²).

### Research Q3: UA food production potential

Results indicate that UA using 75% of the available urban land could supply up to 20.4% of population food needs (Table 3). Proportionately less of the population could be fed by UA in lesser land use scenarios (e.g., 50%, 25% of the micro- and meso-UA areas), or lower yield scenarios (nuclear winter, necessitating frost-resistant crops),

### Research Q4: Near-urban land requirement

This scenario where UA supplies food to feed 20.4% of the population, would necessitate at least 1140 ha of near-urban cultivation (industrial cropping adjacent to the city) to make up the shortfall (Table 4), equivalent to 33% the size of the built urban area.

This near-urban industrial cropping would require at least 134,468 L of liquid fuel for farm machinery per annum. If external supply of liquid fuel was not possible, then liquid fuel could be supplied by cultivating 110 ha of canola seed as a feedstock to produce biodiesel (Table 5). Proportionately more near-urban land and biofuel feedstock would be required in the scenarios where proportionately less UA land area was cultivated.

Fig 3 illustrates schematically the minimum land area requirements for feeding the city. The total additional near-urban area required (combining both food crop and biofuel feedstock area) is equivalent to 39.5% of the total land area of the built urban environment.

### Exploratory results

If looking at data from experienced permaculture growers [27], rather than the meta-analysis of Payen et al., then Jerusalem artichokes (sunchokes) and peas are high yield in a no fertiliser/ no pesticide scenario, especially if growing artichoke as perennial and harvesting tubers, requiring just 113 m² of area to fed a person per year (compared to the above 292 m² for UA peas and 156 m² for near-urban industrial potatoes). We provide direct comparison between the meta-analysis and results from these permaculture growers for the other less efficient crops in Supporting Information Table S3 in S1 Text. However, most city residents will not be experienced permaculture growers, so we consider this finding based on a single detailed report to be a sensitivity analysis, not the main result.

Table 3. Potential for UA to meet combined protein and food energy needs of the palmerston North population.

| Crop scenario* | Number of people fed (% of population) | | |
| --- | --- | --- | --- |
| | 75% of potential UA area used | 50% | 25% |
| Peas only | 18,751 (20.4%) | 12,501 (13.6%) | 6250 (6.8%) |
| Sugar beet & spinach | 15,870 (17.3%) | 10,580 (11.5%) | 5290 (5.8%) |
| Sugar beet & spinach (Yield as per a sunlight reducing and precipitation reducing 5 Tg nuclear winter) | 14,617 (15.9%) | 9744 (10.6%) | 4872 (5.3%) |
| Turnips & chicory (Yield as per 150 Tg nuclear winter) | 2921 (3.2%) | 1947 (2.1%) | 974 (1.1%) |

* Because of significant uncertainty, the table reports results ranging from the most efficient crops in a no winter (normal yields) scenario with intensive UA (75% of UA space), through to the next most efficient crops in a severe winter scenario with 25% of potential space used for UA.

**Table 4. Additional near-urban land area (ha) required for industrial cropping to feed remainder of population (i.e., in addition to UA detailed in** Table 3**).**

| Crop scenario | Additional near-urban land area that needs to be cultivated | | |
|---|---|---|---|
| | 75% of potential UA area used | 50% | 25% |
| Peas (UA) + Potatoes (near-urban) | 1140 | 1237 | 1335 |
| Sugar beet & spinach (UA) + Wheat & carrot (near-urban) | 1716 | 1836 | 1955 |
| Sugar beet & spinach (UA) + Wheat & carrot (near-urban) (Yield as per 5 Tg winter) | 1894 | 2014 | 2133 |
| Turnip & chicory (UA) + Sugar beet (near-urban) (Yield as per 150 Tg winter) | 6274 | 6343 | 6411 |

**Table 5. Diesel fuel (L) needed for near-urban industrial agricultural machinery (ha of canola to supply it as biodiesel)\*.**

| Crop scenario | Volume of liquid fuel required (ha of canola feedstock to supply it) | | |
|---|---|---|---|
| | 75% of potential UA area used | 50% | 25% |
| Peas (UA) + Potatoes (near-urban) | 134,468 L (110 ha)* | 145,974 L (119 ha) | 157,480 L (128 ha) |
| Sugar beet & spinach (UA) + Wheat & carrot (near-urban) | 73,788 L (60 ha) | 78,929 L (64 ha) | 84,070 L (68 ha) |
| Sugar beet & spinach (UA) + Wheat & carrot (near-urban) (Yield as per 5 Tg winter) | 81,441 L (72 ha) | 86,581 L (77 ha) | 91,722 L (81 ha) |
| Turnip & chicory (UA) + Sugar beet (near-urban) (Yield as per 150 Tg winter) ** | 740,313 L (1542 ha) | 748,422 L (1559 ha) | 756,531 L (1576 ha) |

*An additional 5–15% of canola feedstock area is required for biodiesel to produce the canola itself, as detailed for the NZ setting [11].

**Liquid fuel required in this scenario is much greater because UA crop is not the most efficient, UA yield is reduced in 150 Tg nuclear winter, near-urban crop not optimal, near-urban yield reduced in 150 Tg winter, 'potato' diesel requirement per ha applied for sugar beet is greater than 'wheat' requirement, land area of canola needed increased due to 150 Tg winter.

## Discussion

### Urban and near-urban agriculture potential

With the potential to feed up to 20.4% of the population, we found there is insufficient micro- and meso- UA potential within the bounds of a suburban-style globally median-sized city to feed the city after an abrupt global catastrophe, even when public spaces and residential yards are optimally prepared and planted with the most productive UA crops. A modest amount of near-urban land is additionally required for industrial cropping to produce food along with biofuel to run the machinery for industrial agriculture.

Situating our results in the literature, we note that most previous studies we identified (some of which are listed in Table 1), examined only fruit and vegetable intake needs, and not food energy or protein. McDougall et al.'s 2020 study of urban agriculture potential in Sydney, Australia, was an exception. These authors reported that UA could supply calories feeding 33.8% of the population (compared to our finding of 20.4%), however, as we have noted McDougall et al. probably overestimate supply due to their method of calculating potential area for UA (which includes paved yard area by default). Munya's study of Auckland, New Zealand, also considered overall diet, finding that 25% of yard area in low-density suburbs could contribute 9.4–12.3% of diet. This implies that 75% of yard use could supply 28–37%, which is higher than

## A food security strategy for a median-sized city in the normal climate scenario

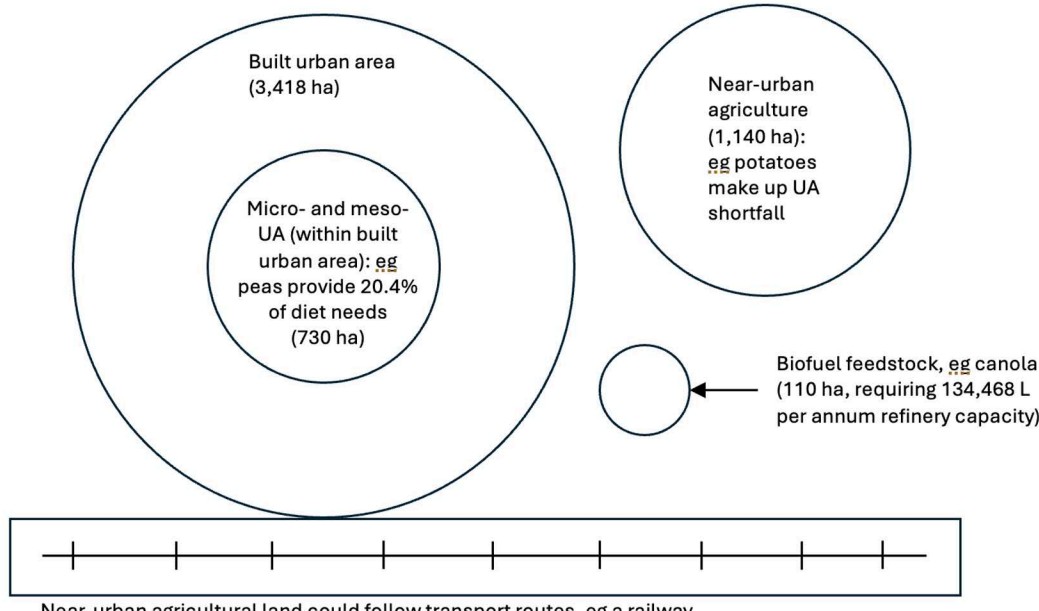

**Fig 3. Land required for food security for a median-sized city.** The figure visually depicts the food security land area footprint of the city. Circles illustrate relative scale of land available for micro-, and meso-urban agriculture (UA) and the population this land area could feed. Also represented is the additional land area of near-urban industrial cropping (including the required biofuel feedstock production), that could ensure food security (dietary energy and protein) for the population of a median-sized city (in this case Palmerston North, New Zealand).

our finding, but in a lower density residential area, emphasizing the value of each city conducting a local analysis, potentially using the simple method we have described.

If the most efficient crops are chosen (peas in the UA space and potatoes on near-urban land) then an additional area of 1250 ha near-urban land must be cropped in a normal climate scenario, even if trade in global fuels ceases (1250 ha is the sum of food crop area in Table 4 and biofuel feedstock area in Table 5). This production would ensure sufficient food energy and protein for the city population as well as biofuel to run the agricultural machinery. It is important to note that less than 9% of this land (110 ha) is required for the biofuel feedstock production. When considering only a minimal amount of biofuel to run agricultural machinery, the argument that biofuel feedstock would displace food for humans does not apply. In fact, sowing this small percentage of land with a crop like canola maximizes food production (i.e., by allowing industrial machinery to operate) in no-fuel-trade scenarios. In normal times the canola can be used to produce food oil. We have further addressed the criticisms of biofuel production and various trade-offs in previous work [11].

The total area of 1250 ha is equal to 39.5% of the built urban environment. More land would be required for less efficient crops, or in the nuclear winter scenarios where frost resistant crops such as wheat, carrots, sugar beet and spinach would be optimal. This near-urban cropping could be configured to take advantage of the most fertile soils, transport routes such as electric railways or rivers, or other features of local geography. Larger denser cities which have high-rise apartments would require even more near-urban land. All this cultivation could be encouraged through judicious land zoning policy in advance of a catastrophe.

To meet the liquid fuel needs, in the base scenario, refining capacity for at least 134,000 L of biodiesel per annum would be required. This volume is equal to only 0.05% of New Zealand's

business as usual national consumption of 295 million L of diesel for off-road agricultural purposes and is less than 1% of the annual output capacity of one former New Zealand canola seed biodiesel refinery [11]. In the 5 Tg nuclear winter scenario 81–92,000 L is needed. The requirement for biodiesel is less in the winter scenario because the frost resistant crops wheat and carrot require less fuel to produce than potatoes, but require more growing land. Some cities will require more fuel for transport if they are not surrounded by arable land, meaning that 'near-urban' farming must be some distance away. Electric drying and milling capacity for wheat, if grown, would be needed and the wheat would need to be stored appropriately to ensure year-round supply. Anticipation of these needs is required, given the high versatility and storage potential of wheat.

Although other studies of UA have noted high potential for food production [35], in our analysis of optimized crops the near-urban yields were higher than micro-/meso-UA yields. This finding may reflect the temperate climate in this case study country setting and the high efficiency of its industrial cropping sector relative to UA. Furthermore, near-urban cropping involves vastly fewer hours of human labor to feed the population than scaling up UA.

That said, prudent GCR planning should assume some risk of break-down in industrial cropping and should anticipate the need for scaling up micro-/meso-UA. Key benefits of including micro-/meso-UA in the mix include reduced transport needs and conservation of liquid fuel, the ability to grow crops with high yields by mass and with high water content, where freshness is needed, requiring minimal capital-intensive processing, as well as diversifying the range of foods available, to meet nutrient needs, and psychological reassurance for urbanites concerned with major levels of social collapse. Therefore, UA cultivation is useful in addition to producing optimized crops in the near-urban space for survival.

UA does not feature in many cities' urban plans, yet an integrated food and energy strategy, based on simple calculations of minimum requirements, could help ensure food supply in an extended global catastrophe. The focus of any strategy should be on facilitating and optimizing industrial cultivation of near-urban agricultural land, supplemented by the scale-up of UA (including assistive tools such as electric rototillers) and local biofuel and grain processing facilities. We acknowledge the competing demands on near-urban land, including development. This is more reason to carefully plan activities at the city fringe, including a GCR perspective, rather than allowing mere urban sprawl.

Our method is straightforward and could be undertaken for other communities, having contextual understanding of what is possible in their areas. There could be projects to prepare land to be fertile for cropping, or for citizens to learn how to do so if ever required.

Previous studies have quantified UA potential of residential lots. However, many of these overestimate residential potential and few previous UA studies attempted to quantify self-sufficiency by optimizing crop selection or considering the most efficient way to overcome food energy and protein shortfall (see example studies in Table 1). We have considered both factors.

Our results are based on open-air and ground grown vegetables. We excluded hydroponic/ LED/artificial light/indoor/greenhouse/polytunnel production data, thereby avoiding the need for large quantities of materials, which may not be available in a post-catastrophe scenario. Wheat has the advantages that less volume per day needs to be eaten (compared to large volumes of potatoes to meet protein needs), it is frost-resistant, has long shelf-life, it is easier to store the seed than for potatoes, it provides a multipurpose food when milled, and surpluses can be used to feed to poultry for egg production.

## Potential challenges

**Seasonality.** Some critiques of UA are worth discussing further. Weidner et al. note that seasonality issues mean that supply variance and storage are critical considerations

[23]. Additionally, there are logistical issues around processing and packing food. Our optimized near-urban crops, namely potatoes, wheat, and carrot store reasonably well, with wheat offering multi-year storage if correctly managed. Potatoes can also be kept in the ground. The optimal UA crop (peas) can be dried and stored for long periods (at least 6–12 months).

**Processing.** We note the need for processing facilities in our scenarios. A canola biodiesel refinery, and wheat mill are both resilience investments that could be considered by cities.

**Inputs.** Small-scale urban agriculture can result in high yields but requires judicious management of inputs for sustainability [38]. Yield of UA and near-urban agriculture will vary based on soil quality, availability of water, agrichemicals, fertilizer and organics, fuel for machinery, as well as expertise. A stockpile of sufficient seed is required for the initial planting (depending on what is presently grown, and the local/regional seed industry), plus knowledge of how to harvest seed for subsequent cycles. We note however, that the impact of many of these factors is likely modest when compared to the impact of sunlight and precipitation reduction in a 150 Tg nuclear winter scenario. Hence, the 150 Tg calculations in Table 3, offer a guide to extreme worst-case requirements for UA and near-urban agriculture than in more likely scenarios.

**Water.** We assumed continued electricity supply, and the provision of reticulated water for irrigating UA and near-urban crops. However, in dry seasons there may be water restrictions in some areas, potentially at peak growing time. Reservoir capacity could be increased, industrial water use curbed in a catastrophe and reticulated pipes maintained to repair and prevent leaks common in urban water supply. Urban supply can be supplemented with roof rain collection, but rainfall is irregular, crops may require five liters per square meter per day, and previous research has found that large storage tanks may be required for UA [16].

**Fertilizer/Organics.** Land converted to UA may need significant input of fertilizer and/or organic material and any GCR that ended trade in liquid fuel could impact fertilizer supply. Land currently cropped may have enough nutrients at least for the first year. Cities/regions should audit the amount of stored fertilizer, optimize fertilizer use by scaling down less efficient food sources, such as livestock, and aim for more circularity in management of organic waste, for example with municipal composting facilities, or storage of soil from construction sites. Community programs to enhance and prepare soil could help develop expertise. One Australian study calculated that if areas with contaminated soils are avoided, unrecovered organic waste can meet a reasonable proportion of that required to establish garden beds in just a few years [24]. Leguminous crop rotation would help (via nitrogen fixation), along with bagged fertilizer from waste treatment plants. Our exploratory results when examining yield reports of permaculture growers indicate that fertilizer (and agrichemicals below) may not be needed once soil quality is sustained [27].

**Agrichemicals.** The need for agrichemicals such as pesticides could be reduced by crop rotation, companion planting, vertical gardening, and use of integrated pest management. Also, use of disease-resistant strains (e.g., some potato varieties have relatively high blight resistance), including developing genetically modified crops that are both disease-resistant and more suited for nuclear winter scenarios (i.e., mature in a shorter growing season, are cold-resistant and drought-resistant).

**Expertise.** Expert gardeners appear to produce more food per land area than amateurs (see Supporting Material) [27]. With incentives from local government, professional farmers could grow optimized crops in the near-urban space. However, scaling up micro-/meso- UA

depends on appropriate expertise. Pre-catastrophe urban farming projects could allow citizens to engage regularly with farming and help facilitate local organic waste schemes and teach fundamentals. Projects should include pilot studies of optimal methods for converting residential lawn to productive UA spaces, which could be scaled post-catastrophe. This post-catastrophe scaling could be overseen by experienced urban gardeners and permaculture growers. Although current UA gardens fall far short of what is needed for complete food self-sufficiency, increasing UA activism and community gardens could amplify knowledge of how to scale UA during a catastrophe. For reference, UK current allotment provision is only 16% what it was in the 1950s [20].

**Liquid fuel.** Efficient near-urban agriculture requires supply of liquid fuels for farm machinery. Crops such as potatoes are particularly fuel demanding, when farmed industrially, requiring approximately 118 L of diesel/ha [39]. We have previously analyzed the potential for essential liquid fuel to be supplied as biodiesel or renewable diesel in a catastrophe [11]. Growing industrial wheat in the near-urban space would consume 0.97 L per person per annum. This would conserve nearly 6 L of diesel per person per annum if wheat substituted for dairy production post-catastrophe (dairy consumes 6.9 L of on-farm diesel per person per annum for the same food energy provided). Even more would be conserved if the substituted dairy was more distant requiring transportation fuel.

**Contamination.** Much urban soil is contaminated, in part due to leaded gasoline use until around 1980 in developed countries. Use of compost can help mitigate potential contamination (e.g., from microbes and toxins in commercial waste), along with peeling vegetables, and the actual health risk from contamination may not be as high as thought [40], especially when the comparator is potential starvation post-catastrophe.

**Uncertainty.** Many factors are context-dependent, and our calculations should be seen as approximations. UA yield is uncertain, this was overcome to some extent in our analyses by using data from a large global meta-analysis, and limiting data to open-air growing, as well as more reliable commercial yield data for near-urban agriculture. Trade collapse in a GCR is also highly uncertain, we conservatively assumed zero imported fuel. The impact of loss of fertilizer, agrichemicals, or even electricity is less certain and further research is needed. The impact of a nuclear winter on growing climate, or if such a winter occurs at all, is also highly uncertain. We have previously discussed this uncertainty [11], and included a relatively severe nuclear winter scenario (150 Tg soot) to try to avoid underestimating the level of preparation needed. We applied yield reduction estimates for New Zealand [37], but these would need to be adjusted for cities in other countries. We have also assumed that all land is equally productive, but it is difficult and expensive to exploit new land for agricultural use [13], therefore expected yield in any particular context might need an adjustment. This, however, is just more reason to develop pre-catastrophe routine production with near-urban industrial agriculture of optimally resilient crops.

## Policy recommendations

Although we did not set out to develop a set of policy recommendations, some broad policy approaches are immediately apparent. Food production is often divorced from urban planning [41], and there is often discrepancy between urban land with UA potential and land zoned for UA. For UA to provide GCR resilience it must be embraced pre-catastrophe at the policy level.

- Central and local government organizations responsible for long-term resilience planning should include food resilience workstreams, particularly with a view to food system and trade disrupting global catastrophes.

- Municipal authorities should consider zoning plans that encourage cultivation of near-urban land, especially existing cropping land, and manage urban sprawl. Cities are often built where they are because the nearby land is highly productive.

- Near-urban production zones might most optimally be configured along transport routes such as railway lines or navigable waterways thereby linking the most productive near-urban land with processing facilities (e.g., for wheat or biodiesel feedstock).

- Authorities could prioritize robust infrastructure in the near-urban area (renewable electricity grid; water supply for irrigation; rail transport links). Development of specific amenities and utilities could consider UA, for example, potential UA water use needs to be supported by water planning and infrastructure [42].

- Additionally, UA involving industrial systems could be highly productive, e.g., potentially growing fully mature lettuces and chicories in a month, and all year round [35]. Such systems could be supported where it is cost-effective, thereby providing year-round supply.

- Governments could incentivize pilot programs to enhance UA (especially near-urban agriculture) in different sized towns and cities. This could include central government mandates for use of local near-urban food for government funded food provision in institutions (e.g., meals in hospitals, and school lunch programs).

- Current local government regulations could be amended to facilitate UA and to facilitate trade in local agricultural produce, e.g., local government support for farmers' markets.

- Local governments could consider reducing rates/land taxes for land used for UA and near-urban cropping.

## Future research directions

The results above imply the following further research directions:

- Central and local government agencies could replicate this type of study for different city types and geographies and expand its dimensions (e.g., cost-effectiveness analyses relating to different crops) to quantify population survival needs in a global catastrophe.

- Near-urban farming is much more plausible than micro-/meso-UA to supply food needs in a global catastrophe and a geographic analysis and map of needed and available land for each city would be useful.

- Cost-effectiveness analysis should include small local biodiesel refineries and wheat mills, with the analyses accounting for the expected harm of low probability, high impact global catastrophes.

- Future studies could use citizen science approaches to estimate UA yields under various conditions (inputs, expertise, growing methods; school gardens, prison gardens) to help inform the above planning. Additional studies of intensive permaculture setups tended by experienced growers would be useful.

- Research needed includes justifying the best practices at each scale of UA, judicious use of production factors, sources of water, fertilizer use etc. Digital technology could be investigated to help implement/coordinate UA and food distributions across many stakeholders.

## Study limitations

In this study we provided a detailed analysis of UA potential and then estimated the additional near-urban land required to feed the population. We have not analyzed the geography of the

near-urban land to determine the 'where' of near-urban cropping. That would be the next step. The UA and near-urban yields described would depend on sustaining a degree of societal order that prevented theft of crops and maintained in-country trade in gardening/cropping inputs. Another limitation was that the main meta-analysis of UA yields did not include any data from New Zealand, albeit most of the studies were in high-income countries and temperate zone settings. There was often quite large variation in crop yields for industrial agriculture [34], and also for UA, based on region/country, crop variety, and how early/late in the season that planting occurred [35].

Nevertheless, our method excluded the UA potential from polytunnels, green houses, gardening on rooftops and balconies, existing fruit trees, trees with edible leaves, mushrooms in sheds/basements, aquaculture, indoor food production systems (hydroponics/artificial lighting), and from small livestock in sheds fed on kitchen waste etc (poultry, rabbits, etc). Similarly, we also did not consider potential mechanisms for increasing available land for UA such as by removing paving stones or pavement (e.g., ripping up car parks and tennis courts), growing crops that spread over paved areas (e.g., pumpkins), and planting on street berms etc. Ornamental trees could be trimmed or cut to expand access to sunlight for urban crops.

We also assumed only one harvest per crop per year and yet for some crops in certain settings, two harvests might be possible in normal climate conditions, this could greatly increase the contribution of UA or decrease the near-urban land needed. Furthermore, we did not account for higher yields from ongoing partial harvests (e.g., the outer leaves of lettuce, kale, spinach and beet greens). For some root crops (e.g., sugar beet, beetroot) we only considered the nutrition from the roots and not the edibility of the leaves for either human or livestock feed. Similarly, for some crops we only considered the above ground harvest and not the less typically used roots which are also edible (e.g., for chicory). Given all these factors, our analyses probably provide a likely underestimate of the total theoretical potential UA production of a post-catastrophe city.

Conversely, our modeled yields could be poorer due to deficiencies of horticultural inputs, or poor initial soil quality. Other than for the nuclear winter scenario (with reduced sunlight and precipitation), we made the simplifying assumption of current day yields (i.e., which are typically supported using hybrid seed varieties, fertilizer and pesticide inputs). Furthermore, in catastrophe situations some potential UA and near-urban land might also be diverted for non-food uses such as: medicinal plants, recreational drugs including crops for beer making, and firewood crops etc.

From a nutritional perspective, our optimization was simplistically based on just dietary energy and protein (and not other macro- and micro-nutrients). Also, as per previous work [32], we reviewed but did not include food wastage calculations, given that these could potentially be markedly reduced in post-catastrophe situations with the threat of food shortages.

## Conclusions

This study demonstrates that urban agriculture alone cannot ensure food security for a median-sized city during a global catastrophe, but a combined approach of UA and near-urban agriculture offers a viable pathway to resilience. The analysis reveals several key conclusions. First, while UA contributes meaningfully to food security, it must be supplemented by efficient near-urban cultivation to meet population needs. Second, careful crop selection - specifically potatoes in normal conditions or frost-resistant alternatives like wheat and carrots in nuclear winter scenarios - can minimize the land area required while maximizing caloric and protein output. Third, the concern that biofuel production competes with food security is largely unfounded in the food security context, as only 9% of the required near-urban land would need to be dedicated to biofuel feedstock to maintain essential agricultural operations.

These findings point to several critical recommendations for enhancing urban food resilience. Cities should prioritize protecting and zoning near-urban agricultural land, particularly along transport corridors, while simultaneously developing UA infrastructure and expertise through pilot programs and community initiatives. Local governments should consider tax incentives for near-urban farming and invest in essential processing infrastructure such as grain mills and small-scale biodiesel refineries. Additionally, municipalities could develop integrated food and energy strategies that optimize both UA and near-urban agriculture, including provisions for water security and soil enhancement.

Our study's methodology provides a framework for other communities to assess and enhance their food security potential. Future work could focus on detailed geographic analyses of available near-urban land, cost-effectiveness studies of processing infrastructure, and expanded evaluation of crop yields under various conditions. While our analysis focused on minimal requirements for survival, implementing these recommendations would significantly improve cities' resilience to global catastrophes that disrupt food and fuel trade, while providing auxiliary benefits even in non-crisis periods.

## Supporting information

**S1 Text. Data used in the analysis of crop optimization and yield.**
(DOCX)

## Author contributions

**Conceptualization:** Matt Boyd, Nick Wilson.

**Data curation:** Matt Boyd, Nick Wilson.

**Formal analysis:** Matt Boyd, Nick Wilson.

**Investigation:** Matt Boyd, Nick Wilson.

**Methodology:** Matt Boyd, Nick Wilson.

**Project administration:** Matt Boyd, Nick Wilson.

**Resources:** Matt Boyd, Nick Wilson.

**Validation:** Matt Boyd, Nick Wilson.

**Visualization:** Matt Boyd.

**Writing – original draft:** Matt Boyd, Nick Wilson.

**Writing – review & editing:** Matt Boyd, Nick Wilson.

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
