## [Decision Letter · Decision Letter 0]

30 Dec 2024

PONE-D-24-50856Resilience to Abrupt Global Catastrophic Risks Disrupting Trade: Combining Urban and Near-Urban Agriculture in a Quantified Case Study of a Globally Median-Sized CityPLOS ONE

Dear Dr. Boyd,

Thank you for submitting your manuscript to PLOS ONE. After careful consideration, we feel that it has merit but does not fully meet PLOS ONE’s publication criteria as it currently stands. Therefore, we invite you to submit a revised version of the manuscript that addresses the points raised during the review process.

Please submit your revised manuscript by Feb 13 2025 11:59PM. If you will need more time than this to complete your revisions, please reply to this message or contact the journal office at plosone@plos.org . Please include the following items when submitting your revised manuscript:

We look forward to receiving your revised manuscript.

Kind regards,

Susmita Lahiri (Ganguly)

Academic Editor

PLOS ONE

Journal requirements: When submitting your revision, we need you to address these additional requirements. 1. Please ensure that your manuscript meets PLOS ONE's style requirements, including those for file naming. The PLOS ONE style templates can be found at https://journals.plos.org/plosone/s/file?id=wjVg/PLOSOne_formatting_sample_main_body.pdf and https://journals.plos.org/plosone/s/file?id=ba62/PLOSOne_formatting_sample_title_authors_affiliations.pdf. 2. We note that Figure 1 in your submission contain [map/satellite] images which may be copyrighted. All PLOS content is published under the Creative Commons Attribution License (CC BY 4.0), which means that the manuscript, images, and Supporting Information files will be freely available online, and any third party is permitted to access, download, copy, distribute, and use these materials in any way, even commercially, with proper attribution. For these reasons, we cannot publish previously copyrighted maps or satellite images created using proprietary data, such as Google software (Google Maps, Street View, and Earth). For more information, see our copyright guidelines: http://journals.plos.org/plosone/s/licenses-and-copyright. We require you to either (1) present written permission from the copyright holder to publish these figures specifically under the CC BY 4.0 license, or (2) remove the figures from your submission: 1. You may seek permission from the original copyright holder of Figure 1 to publish the content specifically under the CC BY 4.0 license.   We recommend that you contact the original copyright holder with the Content Permission Form (http://journals.plos.org/plosone/s/file?id=7c09/content-permission-form.pdf) and the following text:“I request permission for the open-access journal PLOS ONE to publish XXX under the Creative Commons Attribution License (CCAL) CC BY 4.0 (http://creativecommons.org/licenses/by/4.0/). Please be aware that this license allows unrestricted use and distribution, even commercially, by third parties. Please reply and provide explicit written permission to publish XXX under a CC BY license and complete the attached form.” Please upload the completed Content Permission Form or other proof of granted permissions as an ""Other"" file with your submission. In the figure caption of the copyrighted figure, please include the following text: “Reprinted from [ref] under a CC BY license, with permission from [name of publisher], original copyright [original copyright year].” 2. If you are unable to obtain permission from the original copyright holder to publish these figures under the CC BY 4.0 license or if the copyright holder’s requirements are incompatible with the CC BY 4.0 license, please either i) remove the figure or ii) supply a replacement figure that complies with the CC BY 4.0 license. Please check copyright information on all replacement figures and update the figure caption with source information. If applicable, please specify in the figure caption text when a figure is similar but not identical to the original image and is therefore for illustrative purposes only.The following resources for replacing copyrighted map figures may be helpful: USGS National Map Viewer (public domain): http://viewer.nationalmap.gov/viewer/The Gateway to Astronaut Photography of Earth (public domain): http://eol.jsc.nasa.gov/sseop/clickmap/Maps at the CIA (public domain): https://www.cia.gov/library/publications/the-world-factbook/index.html and https://www.cia.gov/library/publications/cia-maps-publications/index.htmlNASA Earth Observatory (public domain): http://earthobservatory.nasa.gov/Landsat:
http://landsat.visibleearth.nasa.gov/USGS EROS (Earth Resources Observatory and Science (EROS) Center) (public domain): http://eros.usgs.gov/#Natural Earth (public domain): http://www.naturalearthdata.com/

Reviewers' comments:

Reviewer's Responses to Questions

**Comments to the Author**

1. Is the manuscript technically sound, and do the data support the conclusions?

Reviewer #1: Yes

Reviewer #2: Partly

2. Has the statistical analysis been performed appropriately and rigorously? 

Reviewer #1: Yes

Reviewer #2: No

3. Have the authors made all data underlying the findings in their manuscript fully available?

Reviewer #1: Yes

Reviewer #2: Yes

4. Is the manuscript presented in an intelligible fashion and written in standard English?

Reviewer #1: Yes

Reviewer #2: Yes

5. Review Comments to the Author

Reviewer #1: Review Comments: Manuscript #: PONE-D-24-50856

MS Title: Resilience to Abrupt Global Catastrophic Risks Disrupting Trade: Combining Urban and Near-Urban Agriculture in a Quantified Case Study of a Globally Median-Sized City

General comments

The manuscript addresses an increasingly critical issue of food security in the face of abrupt global catastrophic risks (GCRs), a topic of both scientific and societal importance. The study's focus on a globally median-sized city and its integration of multiple factors, including urban, meso-urban, and near-urban agriculture, is innovative and adds value to the existing literature. The use of Google Earth image analysis to quantify available urban land and the optimization of crops for specific scenarios demonstrate a strong methodological framework. It provides actionable insights, such as the area requirements for urban and near-urban agriculture, and recommendations for municipal land-use policies to enhance resilience to GCRs. The study contains experiments conducted appropriately, and the methodologies have been well elaborated. The results of the study have also been well explained and discussed properly. However, there are many weaknesses in the write-up which need to be improved before publication.

Specific comments

Shortcomings in each section are mentioned separately here under:

Title: Appropriate.

Abstract

While detailed, the abstract may overwhelm readers with too much technical information. Consider simplifying it to emphasize the key findings and their significance more succinctly.

Keywords: Appropriate.

Introduction

The introduction is comprehensive but could benefit from better structuring to ensure clarity. The paragraphs introduce many related but distinct ideas, which can be overwhelming. Consider breaking the text into smaller, clearly delineated sections.

Terms like "micro-UA," "meso-UA," and "near-urban agriculture" are introduced but used interchangeably with other terms like "peri-urban agriculture." Ensure consistent terminology to avoid confusion.

The transition from the introduction to the methods section could be improved. Consider ending the introduction with a clear summary of the study's aims, framing how the methods will address the identified gaps.

Methods

Study Site Description

Consider briefly explaining why this specific population size or climate is particularly relevant to the study's goals. Clarify if the selection of the city impacts the generalizability of the findings to other regions with different population sizes or climates.

Optimization of Crops for Urban Agriculture (UA)

The assumption of one crop cycle per year might be limiting, acknowledge this and consider discussing potential implications in scenarios where multiple crop cycles are feasible. Provide a rationale for excluding certain crops (e.g., herbs, garlic, cereals) from the micro-/meso-UA analysis. This will help readers understand any potential biases or limitations.

Land Availability Analysis

Including examples or supplementary data on inter-rater reliability metrics would strengthen this section. Consider adding more details about the extrapolation method used for scaling sampled residential lot data to the city level. For example, were any socioeconomic or geographic factors considered during extrapolation?

Food Production Estimation

It would be helpful to mention any uncertainties or assumptions in the yield data (e.g., variations in soil quality or local climate conditions). Clarify how the nuclear winter scenarios (e.g., 150 Tg vs. 5 Tg soot scenarios) were integrated into the yield calculations. Did they involve specific yield reduction coefficients?

Near-Urban Cropping Needs

The inclusion of biofuel feedstock connection to the core research questions should be more explicit. Briefly justify why biofuel considerations are essential in the context of urban agriculture. Highlight any assumptions or trade-offs made in estimating the diesel and land requirements for near-urban cropping and biofuel production.

Statistical Analysis: Appropriate

Results

Ensure these figures are visually intuitive, with clear labels and legends to aid interpretation of the data, particularly for readers less familiar with the topic. The discussion of fuel requirements and biofuel feedstock is relevant but could benefit from further elaboration on the environmental implications or potential trade-offs.

References to supporting information (e.g., Table S3) are helpful, but the main text should briefly explain why this additional data is significant. Ensure that the results are tied back to the study's overarching goals and research questions. Highlight how these findings advance the understanding of urban agriculture and food security.

Discussion

The text is dense and requires clearer subheadings for accessibility. Divide the discussion into more distinct sections with descriptive subheadings, such as "Urban Agriculture Potential," "Challenges and Limitations," "Policy Recommendations," and "Future Research Directions." This will improve readability and highlight key arguments.

The text provides numerical estimates without sufficient contextualization. Add comparative statements or visual aids (e.g., tables, graphs) to contextualize figures like the 1249 ha of near-urban land or 134,000 L of biodiesel. For example, compare these numbers with the total available land in a typical city or provide percentages relative to current agricultural outputs.

Policy suggestions lack actionable specifics. Provide concrete examples of successful urban or near-urban agricultural policies from similar contexts. For instance, reference case studies where zoning laws or urban farming incentives have been effectively implemented.

The discussion is focused on a specific case study without broader generalizations. Discuss how findings could apply globally, particularly to cities in varying climates or socioeconomic settings. Highlight adaptations required for regions with extreme conditions or limited resources

Conclusion: Appropriately written.

References: Check the reference style as per the journal format.

Reviewer #2: Author needs to do major revision on the justification for the research, methods, findings and discussion. There is no ethic statement. The author needs to provide author statement. The tables are not proper, author should consider reconstructing them for clarity of the paper.

6. PLOS authors have the option to publish the peer review history of their article (what does this mean? ). If published, this will include your full peer review and any attached files.

**Do you want your identity to be public for this peer review?** For information about this choice, including consent withdrawal, please see our Privacy Policy .

Reviewer #1: **Yes: ** Dr. Tajwar Alam

Reviewer #2: No

---

## [Author Response · Author response to Decision Letter 1]

2 Feb 2025

Response to Reviewers

PONE-D-24-50856

Resilience to Abrupt Global Catastrophic Risks Disrupting Trade: Combining Urban and Near-Urban Agriculture in a Quantified Case Study of a Globally Median-Sized City

Editorial Comments

Response: Thank you, we have now checked these guidelines carefully and made appropriate revisions.

Figure 1: Supply a replacement figure, please specify in the figure caption text when a figure is similar but not identical to the original image and is therefore for illustrative purposes only.

Response: We have now substituted the original Figure 1 with a new Figure 1, which was extracted from public aerial imagery under a CC BY 4.0 licence. We state in the caption that the Figure is illustrative of the images used in the study.

Reviewer 1

General comments

The manuscript addresses an increasingly critical issue of food security in the face of abrupt global catastrophic risks (GCRs), a topic of both scientific and societal importance. The study's focus on a globally median-sized city and its integration of multiple factors, including urban, meso-urban, and near-urban agriculture, is innovative and adds value to the existing literature. The use of Google Earth image analysis to quantify available urban land and the optimization of crops for specific scenarios demonstrate a strong methodological framework. It provides actionable insights, such as the area requirements for urban and near-urban agriculture, and recommendations for municipal land-use policies to enhance resilience to GCRs. The study contains experiments conducted appropriately, and the methodologies have been well elaborated. The results of the study have also been well explained and discussed properly. However, there are many weaknesses in the write-up which need to be improved before publication.

Response: Thank you for the generally positive assessment of our work. We have now addressed the weaknesses in the write-up that you mention and identify specific revisions below.

Specific comments

Shortcomings in each section are mentioned separately here under:

Title: Appropriate.

Abstract

While detailed, the abstract may overwhelm readers with too much technical information. Consider simplifying it to emphasize the key findings and their significance more succinctly.

Response: We have made some wording changes and also now rounded the key “20%” figure. With the Abstract having only three numbers we think it is not too technically complex for a typical scientific readership.

Keywords: Appropriate.

Introduction

The introduction is comprehensive but could benefit from better structuring to ensure clarity. The paragraphs introduce many related but distinct ideas, which can be overwhelming. Consider breaking the text into smaller, clearly delineated sections.

Terms like "micro-UA," "meso-UA," and "near-urban agriculture" are introduced but used interchangeably with other terms like "peri-urban agriculture." Ensure consistent terminology to avoid confusion.

The transition from the introduction to the methods section could be improved. Consider ending the introduction with a clear summary of the study's aims, framing how the methods will address the identified gaps.

Response: We have now added additional subheadings in the Introduction. We have also streamlined the use of terms, so that we exclusively refer to ‘near-urban’ rather than peri-urban and have dropped all ‘peri-urban’ usage. We have also reformulated the transition to the Methods section by spelling out the research questions, as recommended by Reviewer 2.

Methods

Study Site Description

Consider briefly explaining why this specific population size or climate is particularly relevant to the study's goals. Clarify if the selection of the city impacts the generalizability of the findings to other regions with different population sizes or climates.

Response: We have now clarified that the case study we describe is demonstrative and the method can, perhaps should, be deployed for other cities, of other sizes, in other climates. We chose the city of Palmerston North for its globally median size, and because most of the studies in the meta-analysis on UA yield were from temperate locations (North America, Europe and the temperate zone of East Asia). The Study Limitations section notes that yields vary by region/country, but that our study site aligns with the high-income and temperate climate data that predominated in the meta-analysis.

Optimization of Crops for Urban Agriculture (UA)

The assumption of one crop cycle per year might be limiting, acknowledge this and consider discussing potential implications in scenarios where multiple crop cycles are feasible. Provide a rationale for excluding certain crops (e.g., herbs, garlic, cereals) from the micro-/meso-UA analysis. This will help readers understand any potential biases or limitations.

Response: Thank you, we mention multiple cycles in the Study Limitations, and have now added a line explicitly stating that “this could greatly increase the contribution of UA or decrease the near-urban land needed”. We have added the following explanation regarding the herb/garlic/cereal exclusions at the UA level:

“The rationale for this was that herbs and garlic are typically used as condiments and unlikely to be a palatable diet on their own; and because cereal crops typically have low yields (<1 kg/m2) in UA settings according to Payen et al and are more complex to process than vegetables and fruits.”

Land Availability Analysis

Including examples or supplementary data on inter-rater reliability metrics would strengthen this section. Consider adding more details about the extrapolation method used for scaling sampled residential lot data to the city level. For example, were any socioeconomic or geographic factors considered during extrapolation?

Response: Thank you, we do report the inter-rater reliability in the later section on ‘Statistical Considerations’ and have now additionally added the ICC statistic for the micro-UA judgments. We now mention that the extrapolation was simple multiplication of mean lot area times number of lots in the city. No socioeconomic or geographic factors were included.

Food Production Estimation

It would be helpful to mention any uncertainties or assumptions in the yield data (e.g., variations in soil quality or local climate conditions). Clarify how the nuclear winter scenarios (e.g., 150 Tg vs. 5 Tg soot scenarios) were integrated into the yield calculations. Did they involve specific yield reduction coefficients?

Response: We now explicitly state that these uncertainties are mentioned in the Discussion. We also now specify the yield coefficients for the nuclear winter scenarios in the text.

Near-Urban Cropping Needs

The inclusion of biofuel feedstock connection to the core research questions should be more explicit. Briefly justify why biofuel considerations are essential in the context of urban agriculture. Highlight any assumptions or trade-offs made in estimating the diesel and land requirements for near-urban cropping and biofuel production.

Response: We have now provided more explicit details about the relevance and methods associated with the biofuel considerations in this section.

Statistical Analysis: Appropriate

Results

Ensure these figures are visually intuitive, with clear labels and legends to aid interpretation of the data, particularly for readers less familiar with the topic. The discussion of fuel requirements and biofuel feedstock is relevant but could benefit from further elaboration on the environmental implications or potential trade-offs.

References to supporting information (e.g., Table S3) are helpful, but the main text should briefly explain why this additional data is significant. Ensure that the results are tied back to the study's overarching goals and research questions. Highlight how these findings advance the understanding of urban agriculture and food security.

Response: We have now provided clearer, more detailed figure legends to help the reader. We’ve added text that briefly notes what the Supporting Material shows and why. We have added additional text about the usual trade-off between food vs fuel production, and how it does not apply in these GCR scenarios requiring optimal responses (and refer the reader to an earlier study where we discuss this in more detail). We also now reiterate this point in the Conclusion. How these findings advance the understanding of UA and food security is addressed in the Discussion (see responses below to Reviewer 2 – we now situate the findings among the studies surveyed in Table 1).

Discussion

The text is dense and requires clearer subheadings for accessibility. Divide the discussion into more distinct sections with descriptive subheadings, such as "Urban Agriculture Potential," "Challenges and Limitations," "Policy Recommendations," and "Future Research Directions." This will improve readability and highlight key arguments.

The text provides numerical estimates without sufficient contextualization. Add comparative statements or visual aids (e.g., tables, graphs) to contextualize figures like the 1249 ha of near-urban land or 134,000 L of biodiesel. For example, compare these numbers with the total available land in a typical city or provide percentages relative to current agricultural outputs.

Policy suggestions lack actionable specifics. Provide concrete examples of successful urban or near-urban agricultural policies from similar contexts. For instance, reference case studies where zoning laws or urban farming incentives have been effectively implemented.

The discussion is focused on a specific case study without broader generalizations. Discuss how findings could apply globally, particularly to cities in varying climates or socioeconomic settings. Highlight adaptations required for regions with extreme conditions or limited resources.

Response: We have now revised the sub-headings in the Discussion – thank you. We now provide more comparative figures against key results. Of note, we state that the key figure ‘1250 ha’ is equivalent to 39.5% of the total built urban land area of the city, and we illustrate the relationships between these land areas in Figure 3 with representative shapes that are to scale. With respect to specific actionable policies, this is not intended to be a policy paper, rather it is a demonstration of how a city might go about establishing the footprint of its bare minimum food and fuel needs for survival. However, we now explicitly include a policy recommendations section. The key policy recommendation is that food security and such analysis and thinking should be part of city long-term catastrophe resilience planning. We have now made this clear. In general, there is a lack of policy around UA, so it is difficult to find catastrophe and UA resilience case studies.

Conclusion: Appropriately written.

References: Check the reference style as per the journal format.

Response: We have made a detailed check of the journal guidelines and amended style accordingly.

Reviewer 2

Author needs to do major revision on the justification for the research, methods, findings and discussion. There is no ethic statement. The author needs to provide author statement. The tables are not proper, author should consider reconstructing them for clarity of the paper.

Response: Thank you for the feedback, we have now made revisions as per our itemized responses below.

Topic

Resilience to Abrupt Global Catastrophic Risks Disrupting Trade:

Combining Urban and Near-Urban Agriculture in a Quantified Case Study

Of a Globally Median-Sized City.

The topic is too wordy, consider revising it to make it simpler.

Response: Thank you for this suggestion. However, Reviewer 1 found the title to be ‘Appropriate’ (see above), so we have left it unchanged. Nevertheless, we are happy to follow any advice from the Editor.

Abstract

1. The study calculated land space for urban agriculture (UA) using Google Earth Image analysis however, the author failed to state the method used to determine the optimum food crops by dietary calorie energy (kJ) and protein level.

2. Revise and indicate all the methods.

Response: We have revised the methods in the abstract, which now mentions the mathematical optimization for food energy and protein per land area.

I. Introduction

1. The author's justification for the paper is not enough, particularly the weak link of UA to global catastrophe risk (GCR).

2. Again, the author failed to pose relevant research questions to direct the focus of the study. Rather, the questions are under the methods section (see 2.2, 2.3…).

3. Research questions must come before the objective of the study

4. I suggest you revise the introduction

Response: We have now added two citations in the first paragraph supporting the connection between both GCR and food production, and GCR and food supply chains. The paper explores the link between increasing urban food production and decreasing susceptibility to food shortage. Reviewer 1 noted that “The study's focus on a globally median-sized city and its integration of multiple factors, including urban, meso-urban, and near-urban agriculture, is innovative and adds value to the existing literature.” We have revised the study aims section in the Introduction to express the Research Questions.

Methods

2.2 What are the optimal crops to grow in the UA and near-urban spaces?

1. What is the theory behind using linear programming with an Excel solver in a study such as this? The author must explain the rationale for using the method relative to other methods in the literature.

2. In lines 189 - 191, the author “considered crop yield (kg/tonne of harvested crop per land area) and provision of dietary energy (kJ) and dietary protein at levels to feed an average New Zealander” Where did the author pick the data from? Who cultivates the crops, the farmer or the author? The explanation is not clear.

3. How was the data obtained for the nutrient level analysis? Was it a laboratory analysis? Explain.

4. Check line 198. “where at least two study results were reported in the meta-analysis and…..”The sentence is incomplete and not clear

Response: Thank you for these suggestions. We have now added a further justification for the linear programming method. We have clarified that this data comes from previously published studies, and is based on published industrial crop yield data, and New Zealand specific nutrition survey research (citation [33] given at the end of this description provides details). The incomplete sentence has been corrected, a period was changed to a comma and now it makes sense.

2.3 How much land is available for UA in this median-sized case-study city?

1. Using orthorectified Google Earth Images to determine the land available for UA is satisfactory.

Response: Thank you, we agree.

Table 2: Key variables considered in the analysis

The author needs to consider re-constructing Table 2 for clarity.

Response: We have taken another close look at Table 2, and revised some of its labelling. It should now be fairly clear to readers that the table covers a list of the factors considered, their numerical value (or how the value was calculated in the study) and the source of these data.

Results

2.0 Optimized cropping

1. In the methodology, the author's objective was to estimate the optimum yield of dietary energy and protein, but the results focused on only the optimum crop yield (lines 318 – 324). Why did that happen? How were the nutrient levels for the crops analysed?

Response: The nutrient levels (protein and food energy) were obtained from the New Zealand Institute of Plant and Food Research database, this

---

## [Decision Letter · Decision Letter 1]

3 Mar 2025

Resilience to abrupt global catastrophic risks disrupting trade: Combining urban and near-urban agriculture in a quantified case study of a globally median-sized city

PONE-D-24-50856R1

Dear Dr. Boyd,

We’re pleased to inform you that your manuscript has been judged scientifically suitable for publication and will be formally accepted for publication once it meets all outstanding technical requirements.

Kind regards,

Susmita Lahiri (Ganguly)

Academic Editor

PLOS ONE

Additional Editor Comments (optional):

Reviewers' comments:

Reviewer's Responses to Questions

**Comments to the Author**

1. If the authors have adequately addressed your comments raised in a previous round of review and you feel that this manuscript is now acceptable for publication, you may indicate that here to bypass the “Comments to the Author” section, enter your conflict of interest statement in the “Confidential to Editor” section, and submit your "Accept" recommendation.

Reviewer #1: All comments have been addressed

Reviewer #2: All comments have been addressed

2. Is the manuscript technically sound, and do the data support the conclusions?

Reviewer #1: Yes

Reviewer #2: Yes

3. Has the statistical analysis been performed appropriately and rigorously? 

Reviewer #1: Yes

Reviewer #2: Yes

4. Have the authors made all data underlying the findings in their manuscript fully available?

Reviewer #1: Yes

Reviewer #2: Yes

5. Is the manuscript presented in an intelligible fashion and written in standard English?

Reviewer #1: Yes

Reviewer #2: Yes

6. Review Comments to the Author

Reviewer #1: (No Response)

Reviewer #2: The authors have satisfactorily addressed the comments, making the manuscript quite good. the manuscript has improved, the statistical analysis is okay. Again, the manuscript is presented in standard english.

7. PLOS authors have the option to publish the peer review history of their article (what does this mean? ). If published, this will include your full peer review and any attached files.

**Do you want your identity to be public for this peer review?** For information about this choice, including consent withdrawal, please see our Privacy Policy .

Reviewer #1: **Yes: ** Dr. Tajwar Alam

Reviewer #2: No

---

## [Editor Report · Acceptance letter]

PONE-D-24-50856R1

PLOS ONE

Dear Dr. Boyd,

I'm pleased to inform you that your manuscript has been deemed suitable for publication in PLOS ONE. Congratulations! Your manuscript is now being handed over to our production team.

Kind regards,

on behalf of

Dr. Susmita Lahiri (Ganguly)

Academic Editor

PLOS ONE